# The Evolution of Fermented Milks, from Artisanal to Industrial Products: A Critical Review

**Thomas Bintsis** [1,*] and **Photis Papademas** [2]

1    Laboratory of Safety and Quality of Milk and Dairy Products, Faculty of Veterinary Medicine, Aristotle University of Thessaloniki, 54124 Thessaloniki, Greece

2    Department of Agricultural Sciences, Biotechnology and Food Science, Cyprus University of Technology, Limassol 50329, Cyprus

*    Correspondence: tbintsis@vet.auth.gr

**Abstract:** The manufacture of fermented milk products has a long history, and these products were initially produced either from spontaneous fermentation or using a batch of previously produced product, that is, back-slopping. Milk of different mammal species has traditionally been used for the manufacture of fermented milk products. Cow's milk is the basis for most dairy fermented products around the world. Milk from other mammals, including sheep, goat, camel, mare, buffalo, and yak may have been historically more important and remain so in certain regions. The milks from different species have differences in chemical composition and in certain, vital for the fermentation, components. The diversity of fermented milk products is further influenced by the wide variety of manufacturing practices. A great number of fermented dairy products have been traditionally produced worldwide, and many of them are still produced either following the same traditional process or manufactured industrially, using standardized processes under controlled conditions with specified starter cultures. The evolution from traditional to industrial production, their specific regional differences, their special characteristics, and the microbiological aspects of fermented dairy products are discussed. Throughout the evolution of fermented milk products, functional and therapeutic properties have been attributed to certain components and thus, yogurts and fermented milks have gained a significant market share. These products have gained wide global recognition as they meet consumers' expectations for health-promoting and functional foods. The exploitation of microbiological methods based on DNA (or RNA) extraction and recently high-throughput techniques allowed for the accurate identification of the microbiota of fermented milk products. These techniques have revealed the significance of the properties of the autochthonous microbes and provided novel insights into the role of the microbiota in the functional and organoleptic properties of many fermented milk products.

**Keywords:** fermented milk products; fermented dairy products; yogurt; kefir; spontaneous fermentation; back-slopping





## 1. Introduction

Fermentation is probably one of the oldest preservation methods practiced by human beings. Fermentation is the anaerobic catabolism of carbohydrates by microorganisms [1] and fermented foods are defined the foods that are made under controlled, desired microbial growth and enzymatic conversions of their major and minor components [2–4]. The exact origin of the first accidental making of fermented milk products is difficult to establish, but it could date as far as 10,000–15,000 years ago as the way of life of human beings changed from being food gatherers to food producers [5,6]. According to Leonardi et al. [7], animal domestication started in the Middle Euphrates valley around 11,000 B.C. for sheep and goats and around 10,500 B.C. for cows. Archeozoological data around the middle of the 9th millennium B.C., clearly demonstrate the occurrence of changes in the slaughtering profiles

of sheep and goats [8]. It was at that time that the domestication of these animals began; it is most likely that the transition occurred at different times in different parts of the world [9]. Civilizations such as the Sumerians and Babylonians in Mesopotamia, the Pharaohs in northeast Africa, and the Indians in Asia, have been found that were well advanced in agricultural and husbandry methods, and in the production of fermented milks such as yogurt. Neolithic migration from Anatolia to Europe around the late 7th millennium B.C. occurred and the analyses of potsherds have shown that Neolithic nomads transported their dairy subsistence strategy with them [8,10]. In addition, evidence of milk fermentation dates from the Ptolemaic period in Ancient Egypt, where it was depicted on stelae and in hieroglyphics and engravings [11]. The milk was kept in egg-shaped earthenware jars plugged with grass to protect it from insects and was drunk shortly after milking [12].

A fermented milk product from India, called Dahi, was mentioned in about 6000 to 4000 B.C. in the Rig Veda and Upanishad, ancient sacred books of the Hindus [13]. Egyptian tomb murals from 2000 B.C. show fermented dairy products being made, and other murals demonstrate the ancient Egyptian dairy husbandry practices [12]. Descriptions of the cheese-making process from authors such as the Greeks Homer and Aristotle, and Romans Varro and Columella have been reported [14]. Greek cheeses from the islands of Cythnos and Chios in the Aegean Sea became identified by their place of origin in the era of Mycenaean Civilization [8]. According to Persian tradition, Abraham owed his fecundity and longevity to yogurt, and Emperor Francis I of France was said to have been cured of a debilitating illness by consuming yogurt made from goat's milk [9,15]. It is evident that the production of fermented milk products developed in different ways in different areas, probably in response partly to different environmental conditions and partly to different cultural choices of early farmers [10].

Fermented milk products such as sour milk, yogurt, and cheese, evolved throughout the Middle East, Europe, and India [13]. In the hot climate of these areas, the summer temperatures can reach as high as 40 °C, milk turns sour within a short time of milking, and these conditions have probably helped the sour milk to be processed into a viscous fermented milk product, similar to yogurt-like or concentrated yogurts [9]. Gradually, the nomadic tribes evolved certain steps of the fermentation process and managed to bring it under control. This was made, basically, using the same vessels, or with the addition of fresh milk to an on-going fermentation, relying mainly on the indigenous microflora to sour the milk [9].

Although it was evident very early that sour milks and yogurt-like products had enhanced shelf lives, compared with raw milk, these products evolved and gained special importance and popularity due to other properties such as improved nutritional value. Ancient nomads found fermentation to be the best way to alleviate the various symptoms of lactose intolerance [16–18]. With the advancing of genetic modeling based on modern human DNA sampling, it has been shown that humans were universally adult lactose intolerant at that time; it took several thousand years before adult lactose tolerance became widely established in the human population; this occurred for the first time in Central Europe, sometime after the 6th millennium B.C. [8,16,18]. Itan et al. [18] proposed that the trait of lactose persistence emerged about 7500 years ago in the fertile plains of Hungary. It is probable that fermentation, and thus the production of lactose-reduced products, enabled the successful genetic selection for the capacity to express lactose tolerance into adulthood [8].

The aim of this review is to give an overview of the specific characteristics and the microbiology of fermented milk products, excluding cheese, and to study the impact of new culture-independent molecular analyses, employed at present, to the gradual transition, from the spontaneous and back-slopping techniques to the production of standardized industrial products using automated and fully controlled processes.

## 2. Types of Fermented Milks

Milks of various mammal species are used for the manufacture of fermented milk products differ in chemical composition, including significant differences in parameters such as total solids, lactose, fat, protein, and mineral content (Table 1). There is great variation in the chemical composition of milk from the same species and many factors may affect the gross composition of milk; the factors most significantly affecting the processing of milk products are geo-climatic conditions, animal health, breed, feed, season, and the stage of lactation (Figure 1). Cow's milk is the basis for most dairy fermented products around the world. Milk from other mammals, including sheep, goat, camel, mare, buffalo, and yak may have been historically more important and remain so in certain regions [13]. South European countries as well as many Asian, African, and other Mediterranean countries have centuries of tradition in small ruminant farming, such as ewes and goats, and a number of fermented dairy products are manufactured worldwide [19]. Likewise, milk and fermented milk products from domesticated animals such as yak are common dairy commodities in the Himalayas and neighboring regions [20]. There are other mammals used to produce fermented milk products worldwide, for example, the hybrid of domesticated yaks and cows in Kyrgyzstan, Tajikistan, Mongolia, Buryatia, Pakistan, and Tibet [21].

**Table 1.** Mean percentage composition (% *w/w*) of milks from different species that are used for the production of fermented dairy products.

| Species | Total Solids | Fat | Proteins | Lactose | Ash |
|---|---|---|---|---|---|
| Cow | 12.6 | 3.9 | 3.3 | 4.6 | 0.7 |
| Goat | 13.3 | 4.5 | 3.6 | 4.3 | 0.8 |
| Sheep | 18.6 | 7.5 | 5.3 | 4.6 | 1 |
| Buffalo | 17.2 | 7.4 | 3.9 | 4.8 | 0.8 |
| Yak | 17.7 | 6.7 | 5.5 | 4.6 | 0.9 |
| Horse | 11 | 1.7 | 2.5 | 6.2 | 0.5 |
| Donkey | 10.8 | 1.5 | 2 | 6.7 | 0.5 |

After: [22,23].

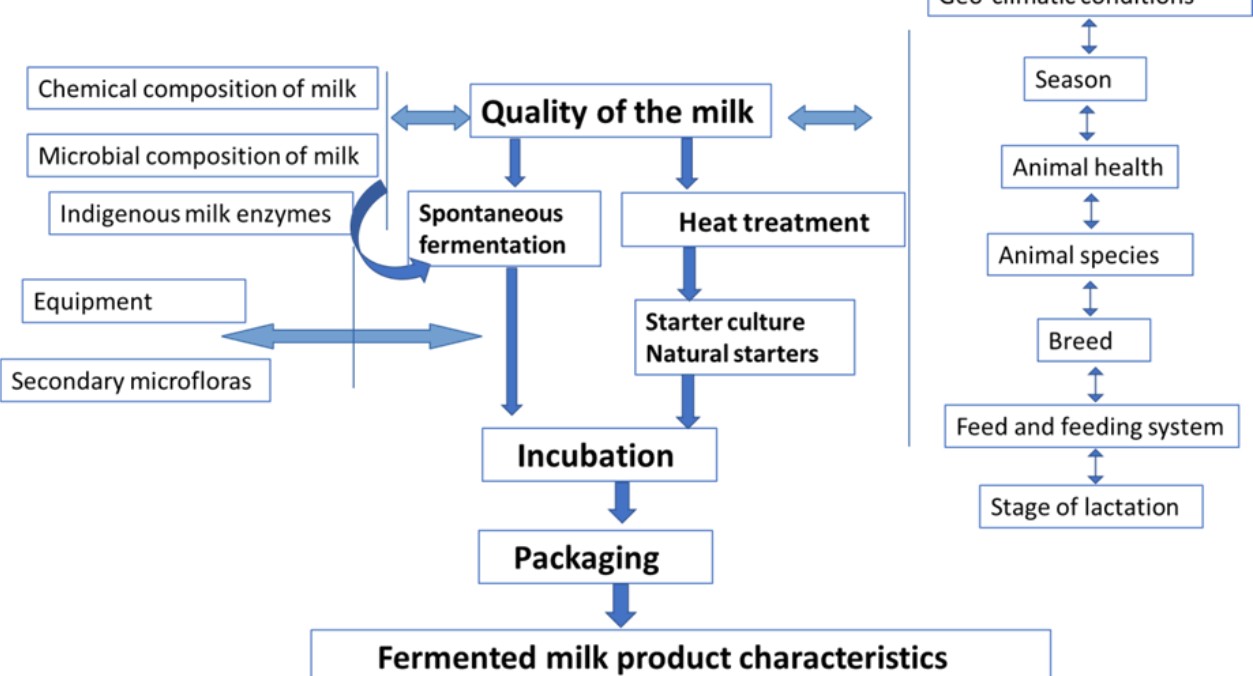

**Figure 1.** Main factors affecting the characteristics of fermented milk products.

Compared to the milk from which they are made, fermented dairy products have unique flavors, textures, appearances, enhanced digestibility, and certain functionalities [5,13].

The variations in milk components, together with the variations in the production processes have created a great diversity of traditional fermented milk products worldwide (Table 2). The variety of manufacturing practices affects the physical, chemical, sensory, and nutritional properties of the product. In addition, processing conditions and product composition also pose strong selection pressure on the microbiota during manufacturing, ripening, and storage [13]. However, products manufactured in different locations still can vary because of microorganisms and culturing practices used in their production. Most of the products shown in Table 2 have persisted over the centuries, even though their production processes have evolved from traditional artisanal manufacture to industrial production. It can be said that the evolution of any given type of fermentation is dependent on the climatic condition of the region so while the thermophilic lactic acid fermentations became predominant in hot and subtropical regions because of the favorable growth conditions of the lactic cultures (40–45 °C), mesophilic fermentations became more popular in colder climates, such as northern Europe [24]. Lactic acid bacteria (LAB) ferment the lactose in milk to produce a fermented product that is pleasant to eat or drink; the latter product is usually referred to as sour milk [9,25]. Fermentation changes the initial characteristics of a food into a product that is significantly different but highly acceptable by consumers.

**Table 2.** Specific and microbiological characteristics of some traditional fermented milk products.

| Type | Product | Continent or Area | Country | Milk | Specific Characteristics | Microorganisms | Form (Drink or Gel) | Reference |
|---|---|---|---|---|---|---|---|---|
| Ic | Acidophilus milk | Europe, America | Greece, Scandinavia, Turkey, Russia, North America | C | Therapeutic milk fermented with species of *Lactobacillus* and *Lactococcus* | *Lb. acidophilus* | Drinkable | [26] |
| Ia | Aewsso or *Huquan* and *Arera* | Africa | Ethiopia | C | Sour milk buttermilk | Mesophilic starters | Drinkable | [27] |
| II | Airag | Asia | China, Mongolia | C, E, Ca | Spontaneous fermentation | LAB and yeasts. Usually *Lb. helveticus, Lb. kefiranofaciens, B. mongoliense, Kl. marxianus Lpb. plantarum, Lb. delbrueckii* subsp. *lactis, Lb. helveticus, Lcb. casei* subsp. *casei, Lcb. casei* subsp. *pseudoplantarum, Lc. lactis, Lc. diacetylactis, Lb. acidophilus, Leuc. mesenteroides, E. faecium, E. faecalis* | Drinkable | [28,29] |
| I | Amasi | Africa | Zimbabwe, Republic of South Africa | C | Spontaneous fermentation | | Viscous | [30,31] |
| II | Ayib | Africa | East and Central Africa | G | Spontaneous fermentation | Mesophilic LAB and yeasts | Drinkable | [32] |
| Ib | Ayran | Asia | Central Asia | C | Yogurt-based beverage | *Lb. delbrueckii* subsp. *bulgaricus, Lb. helveticus, Str. thermophilus* and yeasts | Drinkable | [33] |
| Ib | Ayran | Europe and Middle East | Turkey, Bulgaria | C | Salt-containing yogurt drink, made at 8% total solids | *Lb. delbrueckii* subsp. *Bulgaricus* and *Str. thermophilus* | Drinkable | [34–36] |
| Ib | Bio-yogurt | Europe and America | Various | C | Yogurt produced with the addition of intestinal bacteria, probiotic yogurt | *Lb. delbrueckii* subsp. *bulgaricus* and *Str. thermopilus, Lb. acidophilus, B. bifidum, B. longum, B. breve, P. acidilactici* | Viscous and drinkable | [9,36] |
| Ia | Buttermilk | Europe | Various | C | By-product of butter production | Mesophilic LAB | Drinkable | [37] |
| Ia | Chal | Asia | Iran | Ca | Spontaneous fermentation in a skin bag or a bottle at ambient temperature | Mesophilic LAB | Drinkable | [38] |
| Ib | Chanklish | Middle East | Middle East | C, G, S | Condensed yogurt | *Lb. delbrueckii* subsp. *Bulgaricus* and *Str. thermophilus* | Viscous | [36] |
| Ia | Chhas or Matha | Asia | India (northern parts) | C | Buttermilk, by-product of Desi butter, with addition of ice, salt, sugar, and synthetic flavors. | Mesophilic LAB | Drinkable | [39] |

**Table 2.** *Cont.*

| Type | Product | Continent or Area | Country | Milk | Specific Characteristics | Microorganisms | Form (Drink or Gel) | Reference |
|---|---|---|---|---|---|---|---|---|
| I | Chhu | Asia | Tibet | Y | Spontaneous fermentation, cheese-like product, which is used in soups | *Lb. farciminis, Lvb. brevis, Lb. alimentarius, Lc. lactis* and yeasts as contaminants, *Saccharomycopsis* spp. and *Candida* spp. | Viscous, coagulated. | [40,41] |
| I | Chhurpi | Asia | Tibet, Eastern Himalayas | C | Spontaneous fermentation, soft with crumbly mass used in soups or as a side-dish | *Lb. farciminis, Lcb. paracasei, Lb. bifermentans, Lpb. plantarum, Ltb. curvatus, Lmb. fermentum, Lb. alimentarius, Lb. kefir, Lb. hilgardii, W. confusus, E. faecium, Leuc. mesenteroides* | Viscous | [41] |
| II | Chigee | Asia | Mongolia | E | Spontaneous fermentation, sour and alcoholic taste. | Mesophilic LAB and yeasts | Drinkable | [42] |
| Ia | Dadih | Asia | West Sumatra, Indonesia | B | Spontaneously in a bamboo tube, for two days, at ambient temperature, with natural LAB | Mesophilic LAB | Drinkable | [43] |
| I | Dahi/Dadhi/Dahee | Asia | India, Bangladesh | C, B | Spontaneous fermentation, buttermilk | *Lb. bifermentans, Lb. alimentarius, Lcb. paracasei, Lc. lactis* subsp. *lactis, Lc. lactis* subsp. *cremoris, Str. thermophilus, Lb. delbrueckii* subsp. *bulgaricus, Lb. helveticus, Lb. cremoris, P. pentosaceous, P. acidilactici, W. cibara, W. paramesenteroides, Lb. fermentum, Saccharomycopsis* spp., *Candida* spp. | Drinkable | [44] |
| Ib | Doogh | Asia | Afghanistan, Armenia, Azerbaijan, Iraq, Iran, Syria | C | Yogurt-like drink | *Lb. delbrueckii* subsp. *Bulgaricus* and *Str. thermophilus* | Drinkable | [45] |
| Ib | Ergo | Africa | Ethiopia | C, G | Spontaneous fermentation, yogurt-like | *Lb. delbrueckii* subsp. *Bulgaricus* and *Str. thermophilus* | Viscous | [46] |
| II | Felisouka | Europe | Poland | C | Buttermilk with added sugar, which is fermented with *S. cerevisiae* | Mesophilic LAB and yeasts | Drinkable | [36] |
| Ia | Fillbunke | Europe | Finland | C | Elastic texture | Mesophilic LAB | Drinkable | [37] |
| Ib | Frozen yogurts | Europe | Various | C | Low fat | *Lb. delbrueckii* subsp. *Bulgaricus* and *Str. thermophilus* | Viscous, frozen | [47] |
| I | Gariss | Africa | Sudan | Ca | Spontaneous fermentation, product hung in leather bags made of goat skin | *Lb. paracasei* subsp. *paracasei, Lb. fermentum, Lpb. plantarum, Lc. lactis, Enterococcus* spp. and *Leuconostoc* spp. Yeasts as contaminants | Drinkable | [48] |
| I | Gioddu | Europe | Sardinia | S, G | White color with a creamy consistency and acidic taste | *Lb. delbrueckii* subsp. *bulgaricus* and *Str. thermophilus*, with *Lc. lactis* subsp. *lactis* and *Enterococcus* ssp. | Drinkable | [49,50] |
| II | Kefir | Asia, Europe, and Latin America | Russia, China, Mongolia, Tibet, Turkey, Greece, Italy, Hungary, Brazil, Argentina | C, Ca, B, G, S | Self-carbonated, viscous, with uniform and creamy consistency, slightly alcoholic health-promoting effects | Mixture of bacteria such as *Lactobacillus* spp., *Lactococcus* spp., *Streptococcus* spp., *Leuconostoc* spp., and yeasts such as *Debaryomyces* spp., *Galactomyces* spp., *Issatchenkia* spp., *Kazachstania* spp., *Kluyveromyces* spp., *Pichia* spp., *Saccharomyces* spp., *Torulopsis* spp. *Wickerhamomyces* spp. and *Yarrowia* spp. | Drinkable and viscous | [37,51–54] |
| II | Khoormog | Asia | Mongolia | Ca | Spontaneous fermentation, sour and alcoholic taste | Mesophilic LAB and yeasts | Drinkable | [29] |
| Mi | Kishk | Middle East, Africa | Lebanon, Iraq, Turkey, Egypt | C, G, S, Ca, B | Dried fermented milk/cereal mixture | *Lb. delbrueckii* subsp. *Bulgaricus* and *Str. Thermophilus* | Viscous, dried | [55,56] |
| II | Koumis, Koumiss, Kumys, Kumis, Qumys | Asia and Latin America | Kazakh- stan, China, Colombia | E | Spontaneous fermentation | *Lb. delbrueckii* subsp. *bulgaricus, Lb. salivarius, Lb. buchneri, Lb. helveticus, Lpb. plantarum, Lb. acidophilus, Saccharomyces* spp., | Drinkable | [57,58] |

| Type | Product | Continent or Area | Country | Milk | Specific Characteristics | Microorganisms | Form (Drink or Gel) | Reference |
|---|---|---|---|---|---|---|---|---|
| Ib | Krokmach | Europe | Bulgaria | S | Yogurt-like with a high fat and salt content | *Torulaspora* spp. *Str. thermophilus*, *Lb. delbrueckii* subsp. *bulgaricus*, *Lc. Garviae* subsp. *garviae* and *E. faecalis* | Viscous | [59] |
| I | Kule naoto | Africa | Kenya, Tanzania | C | Spontaneous fermentation, the major daily consumed food of the Maasai community | *Lpd. plantarum*, *Lb. paracasei* subsp. *paracasei*, *Lb. fermentum*, *Lb. acidophilus* | Drinkable | [60,61] |
| Ib | Kurut | Asia and MiddleEast | China, Tibet, Turkey | C, Y | Yogurt-like | *Lb. delbrueckii* subsp. *Bulgaricus* and *Str. thermophilus* | Viscous | [62,63] |
| Ib | Labneh anbaris | Middle East | Middle East (Libanon, Syria, Jordan) | C, G, S | Condensed yogurt, ball shaped with more than 30% total solids, preserved in oil | *Lb. delbrueckii* subsp. *Bulgaricus* and *Str. thermophilus* | Viscous | [36,64,65] |
| Ib | Lassi | Asia | India | C | Buttermilk, refreshing beverage. Sometimes, produced with addition of ice, salt, sugar, and synthetic flavors | Thermophilic LAB, *Lb. acidophilus*, *Str. thermophilus* | Drinkable | [66,67] |
| Ib | Leban or Leben | Middle East | Middle East (Libanon, Syria, Jordan) | C, G, S | Yogurt-like | *Lb. delbrueckii* subsp. *Bulgaricus* and *Str. thermophilus* | Viscous | [68] |
| Ib | Lebneh | Middle East | Middle East (Egypt, Libanon, Saudi Arabia, Syria, Jordan) | C, G, S | Condensed Leben, hung in a cloth bag, whey is draining out | *Lb. delbrueckii* subsp. *Bulgaricus* and *Str. thermophilus* | Viscous | [36,69] |
| Ib | Lebneh anbaris | Middle East | Middle East | C, G, S | Condensed yogurt, mixed with herbs and spices | *Lb. delbrueckii* subsp. *Bulgaricus* and *Str. thermophilus* | Viscous | [36] |
| II | Lengfil | Europe | Sweden | C | Elastic texture with EPS | Mesophilic LAB | | [26] |
| Ia | Mabisi | Africa | Zambia | C | Spontaneous fermentation | Mesophilic LAB | Drinkable | [70,71] |
| II | Maconi | Asia | Caucasian mountains | C | Spontaneous fermentation, kefir-like | Mesophilic LAB and yeasts | Drinkable | [33] |
| II | Matsoni | Europe | Georgia, Armenia | C, S, G, B | Back-slopping fermentation | *Lb. delbruekii* subsp. *lactis*, *Lb. delbrueckii* subsp. *bulgaricus*, *Lb. acidophilus*, *Str. thermophilus*, *Lc. lactis*, *Lc. lactis* subsp. *cremoris*, *G. candidus*, *Saccharomyces*, *Candida*, and other species of yeasts | Viscous | [72] |
| II | Mazun/ Matzoon/ Matsun/ Matsoni/ Madzoon | Asia | Armenia | C, G, B | Spontaneous fermentation, with alcohol | Mesophilic LAB and yeasts | Drinkable | [36] |
| I | Meekiri | Asia | Sri Lanka | B | Back-slopping, yogurt-like | *Lmb. fermentum*, *Ltb. curvatus*, *Lb. acidophilus* and *Lpb. plantarum* | Viscous | [73] |
| Ia | Mish | Africa | Sudan | C | Fermented milk for 1 month by back-slopping, spicy | Mesophilic LAB | Viskous | [74] |
| Ia | Mohi | Asia | Nepal | C | Buttermilk with high acidity used as refreshing beverage | *Lb. alimentarius*, *Lc. lactis*, *Lc. cremoris*, and yeasts as contaminants | Drinkable | [20,41] |
| I | Mursik | Africa | Kenya | C | Spontaneous fermentation | *Lpb. plantarum*, *Lmb. fermentum*, *Lb. brevis*, *Lcb. casei* | Drinkable | [75,76] |
| I | Nono | Africa | Nigeria | C | Spontaneous fermentation | Non-standardized microflora | Drinkable | [77] |
| I | Nunu | Africa | Ghana | C | Spontaneous fermentation, yogurt-like | *Lmb. fermentum*, *Lpb. plantarum*, *Lb. helveticus*, *Leuc. mesenteroides*, *E. faecium*, *E. italicus*, *W. confuse* and yeasts as contaminants | Viscous | [78,79] |
| Ia | Omashikwa | Africa | Namibia | C | Fermented buttermilk | Mesophilic LAB | Drinkable | [80] |
| I | Philu | Asia | Bhutan, Eastern Himalayas | C, Y | Back-slopping fermentation, solid cream | *Lb. bifermentans*, *Lcb. paracasei* subsp. *pseudoplantarum*, *Lb. kefir*, *Lb. hilgardii*, *Lb. alimentarius*, *Lcb. paracasei* subsp. *paracasei*, *Lpb. plantarum*, *Lc. lactis* subsp. *lactis*, *Lc. Lactis* subsp. *cremoris* and *E. faecium* | Viscous | [41] |
| II | Podkvasa | Europe | Bulgaria | S | Spontaneous fermentation, Kefir-like | Mesophilic LAB and yeasts | Drinkable | [36] |
| I | Rayeb | Africa | Tunisia | B | Spontaneous fermentation of raw milk | *Str. thermophilus*, *Lb. bulgaricus*, *Lb. helviticus*, *Lb. acidophilus*, *Lb. delbuerkii*, *Leu. cremoris*, *E. faecium*, *E. durans*, *Str. Acidomonas* and *A. viridans* | Viscous | [81] |

**Table 2.** *Cont.*

| Type | Product | Continent or Area | Country | Milk | Specific Characteristics | Microorganisms | Form (Drink or Gel) | Reference |
|---|---|---|---|---|---|---|---|---|
| II | Rob | Africa | Sudan | C, G, S | Back-slopping fermentation | *Lc. lactis* subsp. *lactis*, *Str. thermophilus*, *Lb. bulgaricus*, *Lb. helveticus*, *Lb. acidophilus* and *Lmb. fermentum*, *Candida kefyr*, *S. cerevisiae*, *S. globosus*, *S. exigus*, *Kl. bulgaricus* and *Kl. lactis.* | Viscous, or diluted with water. | [81–83] |
| Ia | Roub | Africa | Sudan | C | Back-slopping fermentation | Mesophilic LAB | Drinkable | [84] |
| Ib | Ryazhenka | Europe | Russia | C | Back-slopping fermentation | *Lb. delbrueckii* subsp. *Bulgaricus* and *Str. thermophilus* | Drinkable | [85] |
| Ia | Semjölk | Europe | Sweden | C | Homemade product | Mesophilic LAB; e.g., *Lactococcus* spp. and *Leuconostoc* spp. | Drinkable | [37] |
| Ib | Shanklish | Middle East | Lebanon, Syria, Turkey | S, C, G | Concentrated yogurt mixed with spices, preserved in olive oil | *Lb. delbrueckii* subsp. *Bulgaricus* and *Str. thermophilus* | Viscous, concentrated | [86] |
| I | Shubat | Asia and Europe | Kazakhstan, Uzbekistan, Russia, China, Mongolia | Ca | Spontaneous fermentation | *Enterococcus* spp., *Lactobacillus* spp. | Drinkable | [87] |
| Ia | Shyow | Asia | Tibet, Labak, Sikkim, Bhutan | Y | Thick-gel | Mesophilic LAB | Viscous | [41] |
| I | Skyr | Europe | Iceland | C, S | Made from skimmed milk, back-slopping starter and rennin are added. The whey is drained out through a cloth bag. | *Lb. delbrueckii* subsp. *bulgaricus* and *Str. thermophilus* and mesophilic lactobacilli. Yeasts may be present as contaminants. | Viscous | [36,88,89] |
| Ib | Snezhanka | Europe | Bulgaria | C, B, S | Yogurt-like with addition of 6% sugar | *Lb. delbrueckii* subsp. *Bulgaricus* and *Str. thermophilus* | Viscous | [36] |
| Ia | Somar | Asia | Tibet | C, Y | Consumed by Sherpas in the highlands of Himalaya | Mesophilic LAB | Viscous | [20,41] |
| Ib | Stragisto | Europe | Greece | S, C | Concentrated yogurt | *Lb. delbrueckii* subsp. *Bulgaricus* and *Str. thermophilus* | Viscous, concentrated | [9,67] |
| Ib | Strained yogurt | Europe and Middle East | Greece, Iran, Turkey | S, C | Concentrated yogurt | *Lb. delbrueckii* subsp. *Bulgaricus* and *Str. thermophilus* | Viscous, concentrated | [9,67] |
| Ia | Suusac | Africa | Kenya | Ca | Spontaneous fermentation of raw milk in cleaned smoke-treated gourds | Mesophilic LAB | Drinkable | [90] |
| Ib | Suzme yogurt | Middle East | Turkey | S, C | Concentrated yogurt | *Lb. delbrueckii* subsp. *Bulgaricus* and *Str. thermophilus* | Viscous, concentrated | [57] |
| Ib | Tarag | Asia | Mongolia | Y, C, G | Spontaneous fermentation, acidic, sour milk | *Lb. delbrueckii* subsp. *bulgaricus*, *Lb. helveticus*, *Str. ther- mophilus* and yeasts as contaminants | Drinkable | [31,32] |
| Ia | Tätmjölk | Europe | Norway, Sweden, Finland | C | Homemade product | Mesophilic LAB, eg *Lactococcus* spp. and EPS-producing *Leuconostoc* spp. | Drinkable | [37] |
| Ib | Torba yogurt | Middle East | Turkey | C, G, S | Concentrated yogurt, strained in a special cloth bag | *Lb. delbrueckii* subsp. *Bulgaricus* and *Str. thermophilus* | Viscous, concentrated | [91] |
| Mi | Trahanas | Europe | Greece, Cyprus | S, G | The product is produced by mixing the fermented milk with wheat, rolled into balls, and sundried. Consumed after boiling | *Lc. lactis*, *Lc. diacetylactis*, *Leu. cremoris*, *Lb. lactis*, *Lcb. casei*, *Lb. bulgaricus* and *Lb. acidophilus* | Viscous | [92,93] |
| Ib | Tuzlu | Middle East | Turkey | C, G, S | Salted yogurt, boiled for 60 min | *Lb. delbrueckii* subsp. *Bulgaricus* and *Str. thermophilus* | Viscous | [94] |
| II | Viili | Europe | Finland | C | Stringy or ropy texture | *Lc. lactis* subsp. *lactis*, *Lc. Lactis* subsp. *cremoris*, *Lc. Lactis* subsp. *lactis* biovar *diacetylactis*, *Leuc. mesenteroides* subsp. *cremoris* and *G. candidus* | Drinkable | [36,95] |
| Ia | Ymer | Europe | Denmark | C | Concentrated fermented milk product | Mesophilic LAB | Viscous, concentrated | [67] |

**Table 2.** *Cont.*

| Type | Product | Continent or Area | Country | Milk | Specific Characteristics | Microorganisms | Form (Drink or Gel) | Reference |
|---|---|---|---|---|---|---|---|---|
| Ib | Yoghurt/ Yogurt/ Yaort/Your t/Yaourti/Y ahourth/ Yo gur/ Yaghourt | Europe and Middle East | Various | C, G, S | Yogurt | *Lb. delbrueckii* subsp. *Bulgaricus* and *Str. thermophilus* | Viscous | [9] |
| Ib | Zabady | Africa | Egypt | B, C | Yogurt-like, consistency as firm as that of yogurt | Thermophilic LAB | Viscous | [96,97] |
| Mi | Zabuli yellow kashk | Middle East | Iran | C | Traditionally prepared from yogurt, wheat flour, salt, and local aromatic herbs and spices (coriander, cumin, turmeric, dill, garlic) | Species of *Lactobacillus*, *Pediococcus*, and *Streptococcus*. | | [98–100] |
| Ia | Zhentitsa | Asia | East Carpathian Mountains | S (whey) | The product is obtained after the production of Bundz cheese | Mesophilic LAB | Drinkable | [36] |

I: Lactic fermentation, Ia: Mesophilic lactic fermentation, Ib: thermophilic lactic fermentation, Ic: Lactic fermentation—Therapeutic milks, and II: Yeast-lactic fermentation, Mi: Miscellaneous. C: Cow, Ca: Camel, E: Equine, S: Sheep, G: Goat, B: Buffalo, Y: Yak. LAB: Lactic acid bacteria, Lb.: *Lactobacillus*, Lcb.: *Lacticaseibacillus*, Lpb.: *Lactiplantibacillus*, Ltb.: *Latilactobacillus*, Lvb.: *Levilactobacillus*, Lmb.: *Limosilactobacillus*, Str.: *Streptococcus*, Lc.: *Lactococcus*, Leuc.: *Leuconostoc*, E.: *Enterococcus*, B.: *Bifidobacterium*, P.: *Pediococcus*, W.: *Weissella*, A.: *Aeromonas*, Kl.: *Klueveromyces*, S.: *Saccharomyces*, G.: *Galactomyces*.

Different classification schemes have been suggested, and the one proposed by Robinson and Tamime [25] is adopted, with some modifications. Fermented milk products can be divided into two main categories: Type I: lactic fermentations, which includes Ia: mesophilic produced milks, Ib: thermophilic and Ic: therapeutic products and type II: yeast-lactic fermentations (Table 2).

Codex Alimentarius has published standard CXS 243-2003; fermented milks are defined as milk products obtained by fermentation of milk, which may have been manufactured from products obtained from milk with or without compositional modification, by the action of suitable microorganisms and resulting in the reduction of pH with or without coagulation (iso-electric precipitation). These starter microorganisms shall be viable, active, and abundant in the product to the date of minimum durability. If the product is heat treated after fermentation the requirement for viable microorganisms does not apply [101]. The starter cultures, symbiotic cultures of *Streptococcus thermophilus* and *Lactobacillus delbrueckii* subsp. *bulgaricus* for yogurt, cultures of *Str. thermophilus* and any *Lactobacillus* species for alternate culture yogurt, *Lactobacillus acidophilus* for acidophilus milk, starter culture prepared from kefir grains, *Lactobacillus kefiri*, species of the genera *Leuconostoc, Lactococcus* and *Acetobacter* growing in a strong specific relationship for Kefir, and *Lb. delbrueckii* subsp. *bulgaricus* and *Kluyveromyces marxianus* for Kumys. The sum of microorganisms constituting the starter culture should be at a minimum of 107 cfu/g, whereas, where a content claim is made in the labeling that refers to the presence of a specific microorganism that has been added as a supplement to the specific starter culture, these should be at a minimum of 106 cfu/g; the yeasts for Kefir and Kumys should be at a minimum of 104 cfu/g in total [101].

## 3. The Expansion of Fermented Milk Products

The remarkable expansion of fermented milk products started in the early 20th century, after Metchnikoff's proposal [102] that the apparent longevity of the hill tribesmen of Bulgaria was a direct result of their life-long consumption of yogurt inspired an interest in the nutritional characteristics of the product that has never abated [5]. Human gut microbiome research has revealed the link between the gut microbiome and different aspects of human health and diseases, and this finding has necessitated studies on fermented foods and their roles in enhancing the microbiome [103]. Functional and therapeutic yogurts and

fermented milks have reached the markets, since the successful commercial probiotic milk beverage, Yakult, which was launched in 1935 in Japan [104].

Yogurt is one of the most popular fermented milk products worldwide. Originating from the Balkans and the Middle East, it has become a major component of the human diet worldwide [105]. Although homemade yogurt is still produced using the "back-slopping method" worldwide, the growing global attention and the increasing demand, led to the production of yogurt on an industrial scale, with full control of the production procedures and the use of heat-treated milks and starter cultures [106]. Yogurt has a viscosity and a distinctive acidic, sharp flavor [9,107]. Yogurt is produced by the symbiotic growth of *Str. thermophilus* and *Lb. delbrueckii* subsp. *bulgaricus*, both present naturally in milk or added as starter culture at 40–45 °C. *Str. thermophilus* grows faster than *Lb. delbrueckii* subsp. *bulgaricus* and then ferments the lactose in the presence of dissolved oxygen and releases more lactic acid, formic acid, and $CO_2$ from urea, compounds that encourage the growth of *Lb. delbrueckii* subsp. *bulgaricus*. In the presence of formic acid, *Lb. delbrueckii* subsp. *bulgaricus* stimulates *Str. thermophilus* by releasing essential or stimulatory amino acids through its proteolytic system [9,26].

During the first years of industrial production, yogurt had limited acceptability in North Americans and European consumers, since natural yogurt can taste extremely acidic to Western palates, and it was not until the various forms of sweetened and fruit-flavored yogurt went on sale that the market for yogurt really expanded. With innovation in packaging and materials, the concept of stirred fruit yogurt as a pleasant and nutritious snack was the main reason for the forthcoming expansion [5]. Yogurt is manufactured today following a very similar procedure as thousands of years ago and remains the most important fermented milk product. It is presented to the consumer in either a gel form (set type, which is incubated and cooled in the final package) or as a viscous fluid (stirred type, which is incubated in the tank with the coagulum to be broken before cooling and packaging) and more locally as a concentrated product (Table 2). The drinking type yogurt, which is similar to stirred yogurt, has the coagulum broken before cooling, but with more severe agitation. Concentrated yogurt is inoculated and fermented just like stirred yogurt, with the difference that after the breaking of the coagulum, the yogurt is concentrated by boiling off some of the water. These concentrated yogurts are often called strained yogurts or strained fermented milks because of the straining of the whey from the coagulum [63,67]. A special concentrated yogurt is Greek yogurt or Greek-style or Stragisto, which has been strained in a special cloth (tsantila) and thus whey is removed giving a product with 21–23% total solids [67], while Labneh is a famous fermented product from Middle East, strained from the traditional yogurt in a special cloth bag for 10–14 h to remove the whey, and some salt can be added to improve the shelf life [90] (Table 2). Frozen-type yogurt is inoculated and incubated in the same process as stirred yogurt, but the cooling is carried out by pumping through a whipper/chiller/freezer in a process similar to the production of ice cream [9]. From the variety of traditional yogurts, and with increasing success in the global dairy market, novel yogurts and yogurt-like products have entered the markets, for example, frozen yogurts, liquid yogurts, fruit-yogurts, strained yogurts, probiotic yogurts, bio-yogurts, therapeutic yogurts; these have acquired enormous market success.

Kefir, is a viscous, acidic and mildly alcoholic fermented milk, with a refreshing taste, originated from the Caucasus region of Asia; it is produced by natural fermentation or from the inoculation of the kefir grains, that is, back-slopping, in milk [90,107–109]. Kefir grains are composed of an insoluble protein and polysaccharide matrix, gelatinous, and yellowish and vary in size from 0.3 to 3.5 cm in diameter [91]. Kefir has, during the last 10 years, presented an enormous expansion in the global markets. Koumis, Kumis, Kumys, or Qumys is another naturally fermented milk product from the Caucasian area, India, Mongolia, and the Middle East. Similar products such as Chigee and Airag are produced in Mongolia and northwestern China (Table 2), [30,110]. The process is similar to that of Kefir and produces a gray-colored liquid milk, lightly carbonated with a sharp alcoholic and acidic taste [89,111].

Traditional dairy fermentations can be performed either by natural, that is, spontaneous fermentation, or by back-slopping. Both types of fermentation of milk are mediated by LAB, which consume lactose and produce lactic acid [1,112]. The most common dairy LAB include species from four main genera: *Lactococcus, Lactobacillus, Leuconostoc,* and *Pediococcus.* In addition to forming lactic acid, these bacteria also modify other constituents of milk resulting in increased bioavailability of nutrients and enhanced quality [9,113]. In addition, LAB and their metabolic products, mainly bacteriocins, inhibit spoilage and pathogenic microorganisms [114–117].

The main disadvantage of the back-slopping method is that the final product may not always be equally stable in taste and quality, as well as pose a high risk of loss of starter culture activity, for example, by bacteriophages and, as a result, the loss of product [30]. The spontaneous fermentation of milk has been largely displaced by the addition of well-characterized and well-defined starter cultures [118–121]. Dairy cultures consist of selected and well-defined strains of LAB species that are produced in concentrated and stable forms [1]. Their wide availability, ease of use, and consistent properties have made them common even in developing countries [13]. Starter cultures used in milk fermentation include *Lactococcus lactis* subsp. *cremoris, Lc. lactis* subsp. *lactis, Lb. delbrueckii* subsp. *delbrueckii, Lb. delbrueckii* subsp. *lactis, Lb. helveticus, Leuconostoc* spp., and *Str. ther-mophilus;* the main function is the acidification of the medium. However, additional functions are performed such as contribution to the development of texture, flavor, and biochemical changes [120–123].

The majority of commercial fermented milks in the markets are manufactured using a mixture of non-traditional and traditional starter cultures (Table 2). Presently, a great variety of fermented dairy products, based on traditional ones, are manufactured worldwide under controlled conditions with specified starter cultures. Even before sustainability was recognized as an issue in agriculture, fermented dairy products were associated with many of the key elements of sustainable food production [13]. They were manufactured in dairy farms which were self-sufficient with fermented milks being produced and consumed on the location of the farm. Certain characteristics in the production of fermented milks, such as the optimal use of natural and human resources, respect for biodiversity and ecosystems, being environmentally sound, and economically fair and viable, providing the consumer with nutritionally adequate, safe, healthy, and affordable food, meet the requirements of the definition of food sustainability [124]. Raw materials, that is milk from different species, used to make fermented dairy products, were traditionally obtained locally and provided consumers with safe, nutritious, and affordable foods. Fermentation is usually conducted under mild conditions, consuming little energy relative to other forms of food processing, and little waste or by-products are generated [13,124]. In addition, safety is also a global sustainability issue, and for certain autochthonous LAB with antimicrobial properties, the use of protective cultures or the addition of herbs may provide biopreservation to ensure food safety and extended shelf-life at very low cost [125].

Fermented milks are recognized as one of the most popular fermented products due to their extended shelf life and characteristic organoleptic properties, as well as, for their health benefits [126–133]. The functions that have been associated with fermented milk products are schematically shown in Figure 2.

The enhancement of the nutritive value is related to the production of certain vitamins from LAB, as well as the increased essential amino acids in fermented milks [89,134]. Studies on Trachana fermentation showed a significant increase in riboflavin, niacin, pantothenic acid, ascorbic acid, and folic acid contents of the product [135,136]. The inclusion of red pepper as an ingredient in Tarhana increased the α-tocopherol and carotenoid contents and antioxidant activity and improved the fatty acid profile [137,138]. The improvement of sensory characteristics is related to the production of flavor compounds, for example, diacetyl, from LAB has been reported to modify certain milk components resulting in increased bioavailability of nutrients and enhanced quality [113]. The enhanced preservation is achieved with the production of antimicrobial compounds that is organic acids,

hydrogen peroxide, diacetyl, and bacteriocins with antagonistic microbiological properties to suppress the growth of undesirable microbiota [139–143].

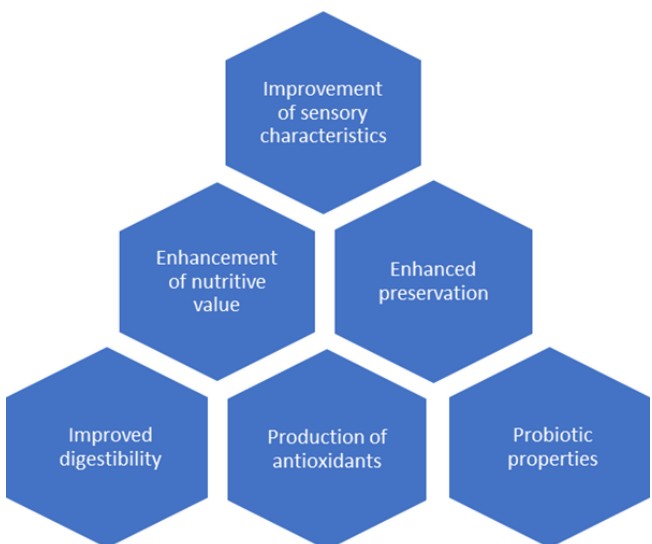

**Figure 2.** Functions associated with fermented milk products.

The improved digestibility of milk by the fermentation process is one of the main health benefits of fermented dairy products; this was probably the main reason for the very early acceptance of fermented milks from lactose-intolerant groups. Lactose intolerance is associated with diarrhea and flatulence induced by lactose metabolites. Because of these, it is nutritionally beneficial to remove lactose; for example, by converting it to lactic acid when fermenting milk, and removing the fraction containing lactose when making fermented dairy products [144]. As a result of the fermentation process conducted using LAB and yeasts, only a little concentration of lactose remains in the final product. Perna et al. showed that the lactose content gradually decreased during storage in yogurt and probiotic yogurt from donkey milk [145].

The production of bioactive compounds, namely conjugated linoleic acid, an anticarcinogenic agent, by *Lb. acidophilus* has been studied [146]. Manzo et al. studied the effects of probiotics and prebiotics, that is, *Bifidobacterium animalis* subsp. *lactis* on conjugated linoleic acid and determined the contents of 10 commercial fermented milk products; they reported that the highest content was observed in fermented milk containing only *Str. thermophilus* and *Lb. delbrueckii subsp. bulgaricus* [147]. Recently, the optimization of certain techniques for maximizing the production of conjugated linoleic acid by *Bifidobacterium animalis* in fermented milk samples was studied [148].

Viili and similar products are made in Sweden, Norway, Denmark, and Iceland [9] and many of these products share a thick and sticky consistency due to the development of extracellular polysaccharide (EPS)-producing strains of *Lactococcus* spp. [149,150]. Besides the technological significance, the health benefits of EPS have also been reported, as they act as nutritional components for colonocytes, liver, and muscle cells, and modulate the host immune system [151]. Antioxidant activity is one of the key functions of the peptides taken from milk proteins and this activity is attributed to the peptides from the proteolysis of casein and to the whey proteins. In addition, *Lb. acidophilus* has been reported to increase the antioxidant activity of yogurt made by autochthonous and commercial starter culture [152].

Probiotics are defined as "live microorganisms which when administered in adequate amounts confer a health benefit to the host" and fermented dairy products are probably the most important food probiotics category; probiotic fermented milks have been extensively studied [128–131,153–159]. Fermented dairy products are generally beneficial in the treatment and prevention of gastrointestinal disease, considering that different LAB strains

show different efficacy across these diseases. Limdi et al. reviewed the therapeutic role of probiotics in gastroenterology and concluded that probiotics appear to have a potential role in the prevention and treatment of various gastrointestinal illnesses, such as irritable bowel syndrome, but it is likely that benefits are species and strains specific [160].

Hypercholesterolemia occurs when there is an elevated level of total cholesterol in the bloodstream and the ingestion of probiotic LAB might be a more natural way to decrease serum cholesterol in humans. Several animal studies have shown that the administration of fermented milks is effective in lowering blood cholesterol levels, although studies in human subjects have shown conflicting results [161].

Observational studies in humans support the importance of the intestinal microbiota in immune development and have found a relationship between probiotics and the development of allergic disease, for example, the treatment of childhood eczema [157].

Clinical evidence has shown that *Lactobacillus rhamnosus* GG can prevent and contribute to the recovery from rotavirus-associated diarrhea in children [162]. Although the mechanism for this protective effect is not clear, it has been shown that *Lb. rhamnosus* GG is able to bind to the mucosal surface of the intestine [155], possibly protecting against intestinal pathogens and associated infections through immunomodulation [163].

Ingestion of probiotic yogurt has been reported to stimulate cytokine production in blood cells and enhance the activities of macrophages [164]. Yakult is a Japanese commercial probiotic milk product that has several health-promoting benefits such as modulation of the immune system, maintenance of gut flora, regulation of bowel habits, alleviation of constipation, and curing of gastrointestinal infections [165,166]. The modulation of the gut microbiota by the administration of *Lactobacillus kefiranofaciens* has been studied in mice [167].

Fermented milks were suggested to have a beneficial effect on cardiometabolic health and especially on type 2 diabetes [168]. Ayyash et al. compared camel to bovine fermented milk and reported in vitro anticancer, antihypertensive, antidiabetic, and antioxidant activities of camel fermented milk [169]. The anti-obesity effect of yogurt fermented by *Lb. plantarum* Q180 in diet-induced obese rats has been studied [170].

During the last 10 years, the functional properties of Kefir were extensively studied [54,108,109,171–173], as well as those of yogurt [174].

## 4. Microbiology of Fermented Milk Products

Another point that has driven the evolution of fermented milk products is the application of culture-independent methods for the identification of microbiota. The microbiota of fermented milk products has been extensively studied using classical microbiology, that is, using culture-dependent methods and phenotypic identification methods. These methods have given significant insights into specific isolates and microbial populations, but the culture media used may not be sufficiently selective for monitoring population dynamics and may fail to recover unculturable bacteria, resulting in an underestimation of microbial diversity [175]. Direct DNA extraction from samples of fermented foods commonly referred to as culture-independent methods is commonly used in food microbiology to profile both cultivable and uncultivable microbial populations from fermented foods [176]. Both culturable and unculturable microbes from any fermented dairy product may be identified using culture-dependent and culture-independent methods; the latter methods had an impact on revealing inter- and intra-species diversity within a particular genus or among genera [177,178]. The most popular culture-independent technique being used in the isolation of microorganisms from fermented foods is a PCR-denaturing gradient gel electrophoresis (PCR-DGGE) analysis to profile bacterial populations [176] and yeast populations in fermented foods [178–180]. Wolfe and Dutton reviewed the microbial communities of fermented foods and concluded that these communities offer a wide range of paradigms for community formation and provide opportunities to understand how to better design synthetic microbial communities for medicine, industry, and agriculture [181]. The omics approaches have contributed to understanding how these microbes affect the organoleptic

properties of fermented dairy products, such as the metabolome and volatilome, and other functional and quality attributes.

Liu et al., 2012 analyzed the bacterial composition of Kurut in Tibet using culture-independent methods, a bacterial 16S rRNA gene clone library containing 460 clones was constructed and the bacterial diversity in Kurut was systematically studied; the authors reported some novel sequences of unknown bacteria [62]. To provide a better understanding of microbial ecology in Kurut, the application of the traditional culture method combined with molecular biology technology would be very useful, and future studies on microbial diversity in traditional fermented dairy products should employ both culture-dependent and culture-independent methods. Watanabe et al. used culture- and molecular biology-based methods to identify 367 LAB strains and 152 yeast strains isolated from Airag and Tarag samples in Mongolia. *Lacticaseibacillus casei*, *Lb. delbrueckii* spp. *bulgaricus*, *Limosilactobacillus fermentum*, *Lb. kefiranofaciens*, *Lactiplantibacillus plantarum*, *Lc. lactis* spp. *cremoris*, and *Str. thermophilus* were the most commonly isolated species [32]. Kochetkova et al. studied the microbiome of more than 50 fermented dairy products from Russia, using culture-independent next-generation sequencing (NGS), and reported that the microbiomes of the same dairy products from different regions were similar in dominant microorganisms and varied mainly in the minor parts of the community [85].

The use of culture-independent methodology has revealed the complex microbiota of kefir grains, which includes a mixture of bacteria such as *Lc. lactis* subsp. *lactis*, *Lc. lactis* subsp. *lactis* biovar. *diacetylactis*, and *Lc. lactis* subsp. *cremoris*, *Lb. kefiranofaciens*, *Lentilactobacillus kefiri*, *Lentilactobacillus parakefiri*, *Lb. helveticus*, *Lb. delbrueckii*, *Lcb. casei*, *Levilactobacillus brevis*, *Lacticaseibacillus paracasei*, *Lpb. plantarum* and *Leuc. mesenteroides*, *Lactobacillus helveticus*, *Leuconostoc citreum*, *Leuconostoc gelidum*, *Leuconostoc kimchi*, *Acetobacter pasteurianus*, and *Acetobacter lovaniensis* [26,182–186], and yeasts such as *Kl. marxianus*, *Saccharomyces cerevisiae*, *Torulopsis kefir*, *Torulaspora delbrueckii*, *Candida kefir*, *Saccharomyces unisporus*, *Pichia fermentans*, *Yarrowia lipolytica*, *Debaryomyces* spp., *Galactomyces* spp., *Issatchenkia* spp., *Kazachstania* spp., *Kluyveromyces* spp., *Pichia* spp., *Saccharomyces* spp., *Wickerhamomyces* spp. and *Yarrowia* spp. [26,187,188]. Kesmen and Kacmaz, using culture-dependent methods identified in Kefir grains and Kefir *Lc. lactis*, *Leuc. mesenteroides* and *Lb. kefiri* as prevalent species, while using PCR-DGGE as a culture-independent method identified *Lb. kefiranofaciens* and *Lc. lactis* as prevalent [189]. Interestingly, Kefir can be made from different milks (Table 2) and the microbiota has been reported to be different from that found in fermented milk as the complex symbiotic interactions between the microbes in the grains differ from those in the milk [190]. The amplicon-based analysis of Kefir from different countries, both in Europe and America, revealed the absence of any clear clustering of the associated microbiomes on the basis of geography; and it was apparent that the populations present in the fermented milk (Kefir) were more homogeneous than the corresponding grains (Kefir grains) from which they were produced [190]. It should be noted that the sequencing data confirmed the change in the species composition and quantitative ratios of the Kefir microbiota with the predominance of lactococci in the final product. Differences between the microbiota of Kefir milk and Kefir grains have also been confirmed by other studies using culture-dependent and culture-independent approaches [191]. Newer identification techniques, like whole metagenome shotgun sequencing, provide more detailed information about the overall microbial structure, in particular for species of low abundance. These methods were able to provide a broader view of the microbial composition and population dynamics of Kefir [107,192–194]. Recently, Alraddadi et al. studied the microbiota of kefir grains and cow's milk kefir, using high-throughput amplicon sequencing; greater diversity in the microbial composition in the kefir than in the kefir grains was found, and the relative abundance of the dominant species, that is *Lb. kefiranofaciens* and *Lc. lactis* and changes over time were observed [194].

## 5. Conclusions

Our ancient ancestors must be acknowledged for providing us with a great variety of fermented dairy products manufactured following a sustainable process. Initially, the primary function of fermenting milk was to extend its shelf life. With this came numerous advantages, such as an improved taste and enhanced digestibility of the milk, and all these reasons resulted in the manufacture of a great number of fermented milk products that consumers are enjoying in different parts of the world. Some of them are still produced using spontaneous fermentation or back-slopping methods; others are produced on an industrial scale using defined starter cultures; many of these products have gained an important place in the global dairy market.

The evolution of fermented milk products has driven, the use of pasteurized milk, more hygienic practices, and the use of defined starter cultures; this has a direct impact on the decreased risks of food safety concerns. On the other hand, the use of heat-treated milk and the addition of defined starter cultures have an impact on the reduction in the diversity of microbiotas. Thus, there is an extended number of studies and efforts to "give back" the lost diversity with the use of adjunct cultures isolated from the autochthonous microbiota. The microbial heterogeneity, with LAB and yeasts derived from the autochthonous microflora of raw milk, manufacturing tools, equipment, and the environment, is an important aspect of fermented milk products.

The microbial heterogeneity has provided several microorganisms with special, functional properties which have been associated with health benefits. As industrially produced fermented milks may lack this microbial diversity, novel starter and adjunct cultures have been developed and are further developing, in order to restore part of the diversity lost by heat treatment and aseptic techniques and produce novel products with improved quality and safety characteristics. The available tools based on NGS technology, and the rise of pioneering integrated multi-omics approaches, have allowed deep understanding and high-resolution analysis of the fermentation process with many novel insights into the fermented milk product microbiome and their role in the organoleptic properties of fermented milk products. Combined multi-omics approaches would facilitate the understanding of the diverse interactions among the autochthonous microbiota, the substrate, and the environment. This diversity can be exploited for the introduction of starter, functional and bioprotective cultures.

As consumers are becoming increasingly aware of the special characteristics and mainly the health-promoting properties of fermented milks, the dairy industry has to face the challenge to manage large-scale production of fermented dairy products, in a sustainable process, without losing the technological and functional characteristics associated with the traditional products from which they are derived.

**Author Contributions:** Conceptualization, T.B. and P.P.; Preparation of the first draft of the manuscript, T.B. and P.P.; Review and editing, T.B. and P.P. All authors have read and agreed to the published version of the manuscript.

**Funding:** This research received no external funding.

**Informed Consent Statement:** Not applicable.

**Data Availability Statement:** Not applicable.

**Conflicts of Interest:** The authors declare no conflict of interest.

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
