# Peer review of "The Evolution of Fermented Milks, from Artisanal to Industrial Products: A Critical Review"

_fermentation, doi:10.3390/fermentation8120679_

Round 1

Reviewer 1 Report

Very interesting and extensive review. I have some minor suggestions indicated in the attached file.

Author Response

Dear Reviewer, thank you very much for the valuable comments made on our manuscript, entitled “The evolution of fermented milks, from artisanal to industrial products: A critical review”, submitted for consideration to be published as a review article in Fermentation.

We have made all corrections suggested by all three Reviewers on the current version of the manuscript, and additionally, we highlighted the corrections from the comments of Reviewer 1 in yellow. No point-to-point reply is included as we made the minor corrections on the manuscript attached.

All authors are aware of and agree to the content of the manuscript. The work of all three Reviewers is acknowledged.  We hope that the present version of the manuscript satisfactorily addresses all the observations made by the Reviewers and meets the quality standard for publication in the Journal of the Fermentation. Thank you very much for your kind consideration.

Yours sincerely,

Thomas Bintsis

Reviewer 2 Report

The present work represents an incursion into the history of fermented dairy products development. Several benefits of consuming dairy products were also shown.

Additional comments were made, mainly regarding English spelling and/or grammar issues.

Author Response

(The authors gave the same response as above.)

Reviewer 3 Report

I want to congratulate you on the excellent work done.

there was a need for a review of this type that spoke of fermented milks.

what is reported in the text is correct and corresponds to my knowledge on fermented milks.

the only notes i can make is:

- write the names of the genres at the end of table 2 using italics;

- line 218 checks the space between the point and the word frozen;

- from lines 260 to 274, check the font of the text;

line 299 checks the space between the point and the word As a ....;

- line 309 verifies the font of the text Bifidobacterium ......;

- line 318 checks the font of the text Lb. acidophilus ......;

- from lines 363 to 377, check the font of the text;

- Check the entire text of the paper for these aspects.

Author Response

(The authors gave the same response as above.)
